# Enhancing Detail Preservation for Customized Text-to-Image Generation: A Regularization-Free Approach

## Abstract

Recent text-to-image generation models have demonstrated impressive capability of generating text-aligned images with high fidelity. However, generating images of novel concepts specified by a reference image remains a challenging task. To address this problem, researchers have been exploring various methods for customizing pre-trained text-to-image generation models. Currently, most existing methods for customizing pre-trained text-to-image generation models involve the use of regularization techniques to prevent over-fitting. Although regularization will ease the challenge of customization and leads to successful content creation with respect to text guidance, it may restrict the model capability, resulting in the loss of detailed information and inferior performance. In this work, we propose ProFusion, a novel framework for customized text-to-image generation, which can tackle the over-fitting problem without the widely used regularization. Specifically, it consists of an encoder network and a novel sampling method. Given a *single user-provided image* from an *arbitrary domain*, the proposed framework can customize a pre-trained text-to-image generation model within half a minute. Empirical results demonstrate that our proposed framework outperforms existing methods.

## 1 Introduction

Text-to-image generation is a research topic that has been explored for years (Xu et al., 2018; Zhu et al., 2019; Zhang et al., 2021; Tao et al., 2021; Zhou et al., 2021), with remarkable progresses recently. Starting from DALL-E (Ramesh et al., 2021) and CogView (Ding et al., 2021), numerous methods have been proposed (Rombach et al., 2021; Ding et al., 2022; Gafni et al., 2022; Ramesh et al., 2022; Saharia et al., 2022; Yu et al., 2022; Zhou et al., 2022; Chang et al., 2023), leading to impressive zero-shot capability in generating text-aligned images of high resolution with exceptional fidelity. Although aforementioned large-scale text-to-image generation models are able to perform text-aligned and creative generation, they may face difficulties in generating novel and unique concepts (Gal et al., 2022) specified by users, especially when the concept is presented via few or only single image. Researchers have exploited different methods in customizing pre-trained text-to-image generation models. For instance, Kumari et al. (2022) and Ruiz et al. (2022) propose to fine-tune the pre-trained generative models with few samples, where different regularization methods are applied to prevent over-fitting. Gal et al. (2022; 2023); Wei et al. (2023) propose to encode the novel concept of user input image in a word embedding, which is obtained by an optimization method or from an encoder network. All these methods lead to customized generation for the novel concept, while satisfying additional requirements described in arbitrary user input text.

Despite these progresses, recent research also makes us suspect that the use of regularization may potentially restrict the capability of customized generation, leading to the information loss of fine-grained details. In this paper, we propose a novel framework called *ProFusion*, which consists of an encoder called ***PromptNet*** and a novel sampling method called ***Fusion Sampling***. Different from previous methods, our ProFusion propose to tackle the potential over-fitting problem by the proposed Fusion Sampling method at inference instead of using regularization, which leads to effective and efficient customization of pre-trained generative models. Our main contributions can be summarized as follows:

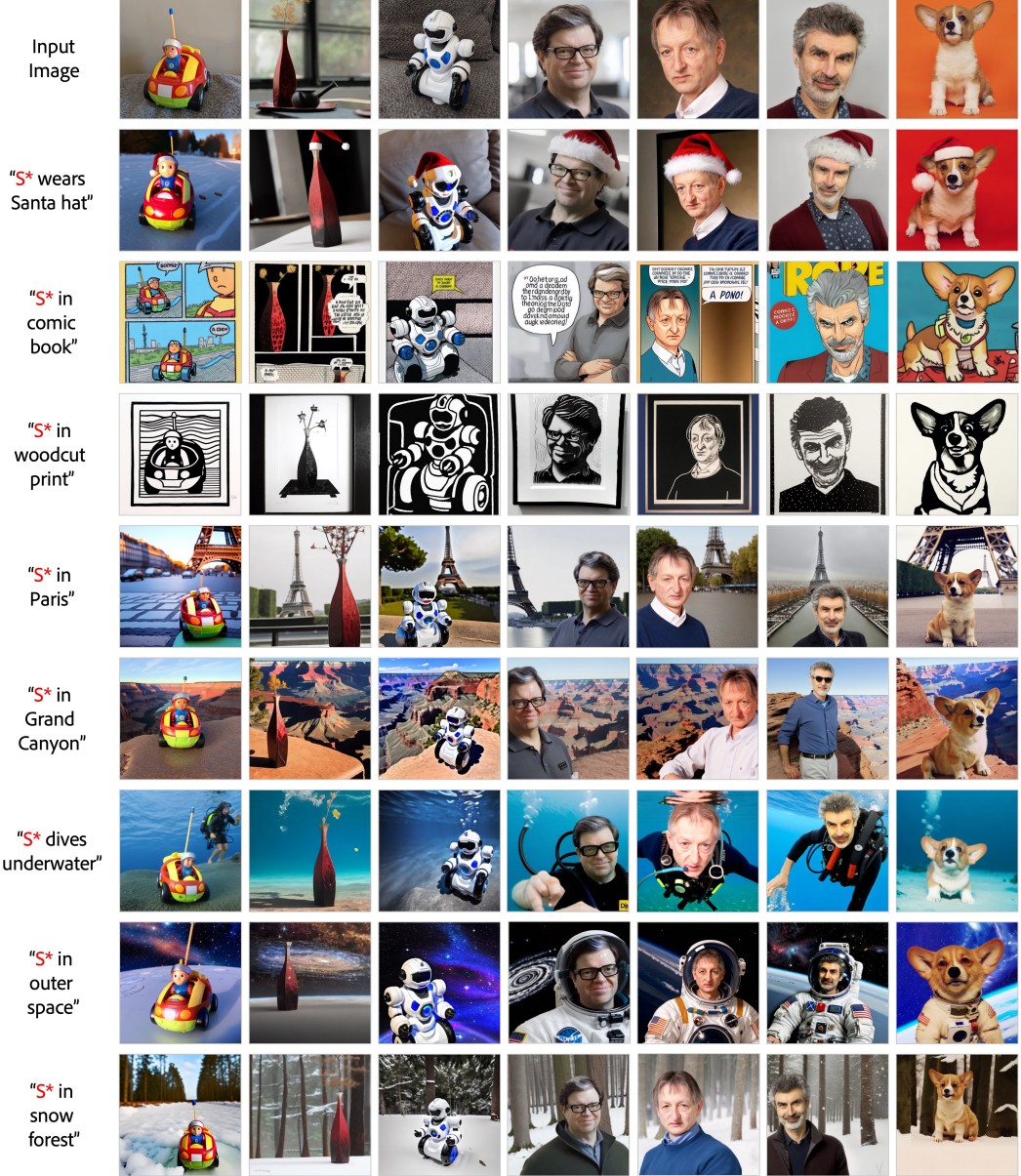

Figure 1: Generated examples from the proposed framework. Given only single testing image from arbitrary downstream domain, ProFusion can perform customized generation in 5 to 25 seconds.

- We propose ProFusion, a novel framework for customized generation. Given a single image containing a unique concept, we can obtain a customized generative model in about 5 to 25 seconds with single GPU. Then the customized model can generate infinite number of creative contents for the unique concept and meets additional requirement specified in arbitrary text;
- The proposed framework does not require any regularization method to prevent over-fitting, which significantly reduces training time as there is no need to tune regularization hyperparameters. The absence of regularization also allows the proposed framework to achieve enhanced preservation of fine-grained details;
- Extensive results, including qualitative, quantitative and human evaluation results, have demonstrated the effectiveness of the proposed ProFusion. Ablation studies are also conducted to better understand the components in the proposed framework;

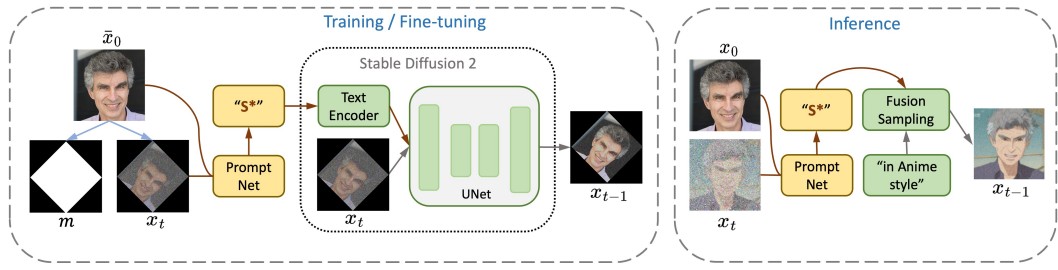

Figure 2: Illustration of the proposed framework. A PromptNet will be trained to output an embedding $S^*$ based on input image. Then Fusion Sampling performs customized generation based on $S^*$ and arbitrary text input.

## 2 METHODOLOGY

Our proposed framework is presented in Figure 2, which consists of a neural network called Prompt-Net and a novel sampling method called Fusion Sampling. PromptNet is an encoder network which infers a text embedding $S^*$ from the reference image $\bar{x}_0$ and current noisy generation $x_t$. Fusion Sampling is a sampling method to generate the desired image conditioned on $S^*$ and arbitrary text.

During training, we augment $x_t$ by random affine transformation, as shown in the figure. Meanwhile, a corresponding mask $m$ can be obtained, which will be used in the training loss for Prompt-Net to make sure that our model only learns the required information.

$$L_{\text{Diffusion}} = \mathbb{E}_{\boldsymbol{x}, \boldsymbol{y}(S^*), t, \epsilon \sim \mathcal{N}(\mathbf{0}, \mathbf{I})} \left[ \| \boldsymbol{m} \odot \{ \epsilon - \epsilon_\theta(\boldsymbol{x}_t, \boldsymbol{y}(S^*), t) \} \|_2^2 \right], \tag{1}$$

where $\boldsymbol{y}(S^*)$ denotes the constructed prompt containing $S^*$, e.g. "A photo of $S^*$", $\odot$ denotes element-wise multiplication. The above training strategy ensures that $S^*$ belongs to the input embedding space of the text encoder from pre-trained Stable Diffusion, which makes it possible for $S^*$ to be combined with arbitrary text for creative generation, *e.g.*, "$S^*$ from a superhero movie screenshot".

Some existing works (Gal et al., 2022; 2023) also try to obtain $S^*$ for customized generation. However, regularization is often applied in these works. For instance, E4T (Gal et al., 2023) proposes to use an encoder to generate $S^*$, which is optimized with

$$L = L_{\text{Diffusion}} + \lambda \| S^* \|_2^2, \tag{2}$$

where the $L_2$ norm of $S^*$ is regularized. Similarly, Textual Inversion (Gal et al., 2022) proposes to directly obtain $S^*$ by solving

$$S^* = \text{argmin}_{S'} L_{\text{Diffusion}} + \lambda \| S' - S \|_2^2$$

with optimization method, where $S$ denotes a coarse embedding[1].

In this work, we argue that although the use of regularization will ease the challenge and enable successful content creation with respect to testing text. It also leads to the loss of detailed information, resulting in inferior performance. To verify this argument, we conduct a simple experiment by training several encoders on the FFHQ dataset (Karras et al., 2019), with different levels of regularization using different $\lambda$ in equation 2. Details of encoder architecture will be provided in experiment section. The resulting models are then evaluated by performing customized generation, as shown in Figure 3. From the figure we can know that:

- Small regularization leads to less information loss, which results in better detail preservation. However, the learned information could be too strong to prevent creative generation with respect to user input text;
- Large regularization leads to successful content creation, while fails to capture fine-grained details of the input image, resulting in unsatisfactory results;

A consequent question is, **is it possible to perform successful customized generation using $S^*$ obtained without regularization so that the details from original image can be well-preserved?** To answer this question, we propose a novel sampling method called Fusion Sampling.

---

[1]For example, $S^*$ is the target embedding for a human face image, $S$ is set as the embedding of the word "face".

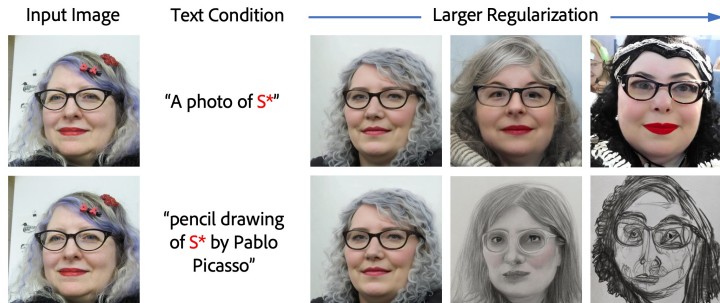

Figure 3: The performance of customized generation is impacted by the level of regularization.

## 2.1 FUSION SAMPLING

Given a PromptNet trained without regularization which can map input image $\bar{x}_0$ into word embedding $S^*$, our goal is to successfully perform customized generation which preserves details of $\bar{x}_0$, and meets the requirements specified in arbitrary prompt containing $S^*$.

The task can be formulated as a conditional generation task with conditions $S^*$ and $C$, where $C$ denotes arbitrary user input text. We start with the most commonly used classifier-free sampling (Ho & Salimans, 2021). To sample $x_{t-1}$ given the current noisy sample $x_t$ and conditions $[S^*, C]$, the diffusion model first outputs the predictions of conditional noise $\epsilon_\theta(x_t, S^*, C)$ and unconditional noise $\epsilon_\theta(x_t)$. Then an updated prediction

$$\tilde{\epsilon}_\theta(x_t, S^*, C) = (1 + \omega)\epsilon_\theta(x_t, S^*, C) - \omega\epsilon_\theta(x_t), \tag{3}$$

will be used in different sampling strategies (Ho et al., 2020; Karras et al.; Song et al., 2020; Song et al.), where $\omega$ is a hyper-parameter for classifier-free guidance.

In previous methods, the reason that vanilla classifier-free sampling does not work is that the information from $S^*$ can become too strong without regularization. As a result, $\epsilon_\theta(x_t, S^*, C)$ will degenerate to $\epsilon_\theta(x_t, S^*)$ and information of $C$ will be lost. Thus, we need to propose a new sampling method to produce a new prediction for $\tilde{\epsilon}_\theta(x_t, S^*, C)$ so that it is enforced to be conditioned on both $S^*$ and $C$.

**Sampling with independent conditions** We begin by assuming that $S^*$ and $C$ are independent. According to Ho & Salimans (2021), we know that

$$\epsilon_\theta(x_t, S^*, C) = -\sqrt{1 - \bar{\alpha}_t}\nabla \log p(x_t | S^*, C), \tag{4}$$

where $\bar{\alpha}_t$ is a hyper-parameter as defined in Ho et al. (2020). By equation 4 and Bayes' rule, we can re-write equation 3 as

$$\tilde{\epsilon}_\theta(x_t, S^*, C) = \epsilon_\theta(x_t) - (1 + \omega)\sqrt{1 - \bar{\alpha}_t}\nabla \log p(S^*, C | x_t). \tag{5}$$

Since we assume that $S^*, C$ are independent, we can further rewrite the above as

$$\tilde{\epsilon}_\theta(x_t, S^*, C) = \epsilon_\theta(x_t) - (1 + \omega)\sqrt{1 - \bar{\alpha}_t}\nabla \log p(S^* | x_t) - (1 + \omega)\sqrt{1 - \bar{\alpha}_t}\nabla \log p(C | x_t)$$
$$= \epsilon_\theta(x_t) + (1 + \omega)\{\epsilon_\theta(x_t, S^*) - \epsilon_\theta(x_t)\} + (1 + \omega)\{\epsilon_\theta(x_t, C) - \epsilon_\theta(x_t)\}.$$

We rewrite it as

$$\tilde{\epsilon}_\theta(x_t, S^*, C) = \epsilon_\theta(x_t) + (1 + \omega_1)\{\epsilon_\theta(x_t, S^*) - \epsilon_\theta(x_t)\} + (1 + \omega_2)\{\epsilon_\theta(x_t, C) - \epsilon_\theta(x_t)\} \tag{6}$$

for more flexibility. Furthermore, equation 6 can also be extended to more complicated scenarios, where a list of conditions $\{S_1^*, S_2^*, ..., S_k^*, C\}$ are provided:

$$\tilde{\epsilon}_\theta(x_t, \{S_i^*\}_{i=1}^k, C) = \epsilon_\theta(x_t) + \sum_{i=1}^k (1 + \omega_i)\{\epsilon_\theta(x_t, S_i^*) - \epsilon_\theta(x_t)\} + (1 + \omega_C)\{\epsilon_\theta(x_t, C) - \epsilon_\theta(x_t)\}.$$

---

**Algorithm 1** Fusion Sampling at Timestep t

1: **Require: Conditions $S^*$ and $C$, noisy sample $x_t$, diffusion model $\epsilon_\theta$, hyper-parameters $0 < \sigma_t, 0 \leq \gamma \leq 1$.**
2: Set $\tilde{x}_t = x_t$
3: // Fusion Stage
4: **for** i = 1, ..., m **do**
5:      Generate $\tilde{\epsilon}_\theta(\tilde{x}_t, \gamma S^*, C)$ by equation 3.
6:      Generate predicted sample $\tilde{x}_0 = \dfrac{\tilde{x}_t - \sqrt{1 - \bar{\alpha}_t}\tilde{\epsilon}_\theta(\tilde{x}_t, \gamma S^*, C)}{\sqrt{\bar{\alpha}_t}}$.
7:      Inject fused information into $\tilde{x}_{t-1}$ by sampling $\tilde{x}_{t-1} \sim q(\tilde{x}_{t-1}|\tilde{x}_t, \tilde{x}_0)$.
8:      **if** Use refinement stage **then**
9:          Inject fused information into $\tilde{x}_t$ by sampling $\tilde{x}_t \sim q(\tilde{x}_t|\tilde{x}_{t-1}, \tilde{x}_0)$.
10:      **else**
11:          Return $x_{t-1} = \tilde{x}_{t-1}$.
12:      **end if**
13: **end for**
14: // Refinement Stage
15: **if** Use refinement stage **then**
16:      Generate $\tilde{\epsilon}_\theta(\tilde{x}_t, S^*, C)$ by equation 6 and perform classifier-free sampling step. Return $x_{t-1}$.
17: **end if**

---

**Fusion Sampling with dependent conditions**     One major drawback of equation 6 is that the independence does not always hold in practice. As we will show in later experiment, assuming that $S^*$ and $C$ are independent can lead to inferior generation.

To solve this problem, we propose Fusion Sampling, which consists of two stages at each timestep $t$: a **fusion stage** which encodes information from both $S^*$ and $C$ into $x_t$ with an updated $\tilde{x}_t$, and a **refinement stage** which predicts $x_{t-1}$ based on Equation equation 6. The proposed algorithm is presented in Algorithm 1. Sampling with independent conditions can be regarded as a special case of Fusion Sampling with $m = 0$. We set $m = 1$ in practice because it works well enough.

The remaining challenge in Algorithm 1 is sampling $\tilde{x}_{t-1} \sim q(\tilde{x}_{t-1}|\tilde{x}_t, \tilde{x}_0)$ and $\tilde{x}_t \sim q(\tilde{x}_t|\tilde{x}_{t-1}, \tilde{x}_0)$. We take the Denoising Diffusion Implicit Models (DDIM) (Song et al., 2020) as an example, while the following derivation can be extended to other diffusion models. Let $\mathbf{I}$ be the identity matrix, $\sigma_t$ denotes a hyper-parameter controlling randomness. In DDIM, we have

$$q(\tilde{x}_t|\tilde{x}_0) = \mathcal{N}(\tilde{x}_t; \sqrt{\bar{\alpha}_t}\tilde{x}_0, (1 - \bar{\alpha}_t)\mathbf{I}) \tag{7}$$

and

$$q(\tilde{x}_{t-1}|\tilde{x}_t, \tilde{x}_0) = \mathcal{N}\left(\tilde{x}_{t-1}; \sqrt{\bar{\alpha}_{t-1}}\tilde{x}_0 + \sqrt{1 - \bar{\alpha}_{t-1} - \sigma_t^2}\frac{\tilde{x}_t - \sqrt{\bar{\alpha}_t}\tilde{x}_0}{\sqrt{1 - \bar{\alpha}_t}}, \sigma_t^2\mathbf{I}\right). \tag{8}$$

By the property of Gaussian distributions (Bishop & Nasrabadi, 2006), we know that

$$q(\tilde{x}_t|\tilde{x}_{t-1}, \tilde{x}_0) = \mathcal{N}(\tilde{x}_t; \mathbf{\Sigma}(A^T L(\tilde{x}_{t-1} - b) + B\boldsymbol{\mu}), \mathbf{\Sigma}) \tag{9}$$

where

$$\mathbf{\Sigma} = \frac{(1 - \bar{\alpha}_t)\sigma_t^2}{1 - \bar{\alpha}_{t-1}}\mathbf{I}, \quad \boldsymbol{\mu} = \sqrt{\bar{\alpha}_t}\tilde{x}_0, \quad b = \sqrt{\bar{\alpha}_{t-1}}\tilde{x}_0 - \frac{\sqrt{\bar{\alpha}_t(1 - \bar{\alpha}_{t-1} - \sigma_t^2)}}{\sqrt{1 - \bar{\alpha}_t}}\tilde{x}_0$$

$$A = \frac{\sqrt{1 - \bar{\alpha}_{t-1} - \sigma_t^2}}{\sqrt{1 - \bar{\alpha}_t}}\mathbf{I}, \quad L = \frac{1}{\sigma_t^2}\mathbf{I}, \quad B = \frac{1}{1 - \bar{\alpha}_t}\mathbf{I}$$

which leads to

$$\tilde{x}_t = \frac{\sqrt{(1 - \bar{\alpha}_t)(1 - \bar{\alpha}_{t-1} - \sigma_t^2)}}{1 - \bar{\alpha}_{t-1}}\tilde{x}_{t-1} + \frac{(1 - \bar{\alpha}_t)\sigma_t^2}{1 - \bar{\alpha}_{t-1}}z$$
$$+ \frac{\tilde{x}_0}{1 - \bar{\alpha}_{t-1}}\{\sqrt{\bar{\alpha}_t}(1 - \bar{\alpha}_{t-1}) - \sqrt{\bar{\alpha}_{t-1}(1 - \bar{\alpha}_t)(1 - \bar{\alpha}_{t-1} - \sigma_t^2)})\}, \quad z \sim \mathcal{N}(\mathbf{0}, \mathbf{I}). \tag{10}$$

With further derivation, we can summarize a single update in our fusion stage as:

$$\tilde{x}_t \leftarrow \tilde{x}_t - \frac{\sigma_t^2\sqrt{1 - \bar{\alpha}_t}}{1 - \bar{\alpha}_{t-1}}\tilde{\epsilon}_\theta(\tilde{x}_t, \gamma S^*, C) + \frac{\sqrt{(1 - \bar{\alpha}_t)(2 - 2\bar{\alpha}_{t-1} - \sigma_t^2)}}{1 - \bar{\alpha}_{t-1}}\sigma_t z, \quad z \sim \mathcal{N}(\mathbf{0}, \mathbf{I}). \tag{11}$$

**Remark 1**   Recall $\tilde{\epsilon}_{\boldsymbol{\theta}}(\tilde{\boldsymbol{x}}_t, \gamma S^*, C) = -\sqrt{1 - \bar{\alpha}_t}\nabla \log \tilde{p}_\omega(\tilde{\boldsymbol{x}}_t | \gamma S^*, C)$, we can rewrite equation 11 as

$$\tilde{\boldsymbol{x}}_t \leftarrow \tilde{\boldsymbol{x}}_t + \frac{\sigma_t^2(1 - \bar{\alpha}_t)}{1 - \bar{\alpha}_{t-1}}\nabla \log \tilde{p}_\omega(\tilde{\boldsymbol{x}}_t | \gamma S^*, C) + \frac{\sqrt{(1 - \bar{\alpha}_t)(2 - 2\bar{\alpha}_{t-1} - \sigma_t^2)}}{1 - \bar{\alpha}_{t-1}}\sigma_t \boldsymbol{z}. \tag{12}$$

From equation 12, we know that our fusion stage is actually a gradient-based optimization method similar to Langevin dynamics (Welling & Teh, 2011). Compared to Langevin dynamics which is

$$\tilde{\boldsymbol{x}}_t \leftarrow \tilde{\boldsymbol{x}}_t + \lambda\nabla \log \tilde{p}_\omega(\tilde{\boldsymbol{x}}_t | \gamma S^*, C) + \sqrt{2\lambda}\boldsymbol{z}. \tag{13}$$

with $\lambda$ being the step size, equation 12 has less randomness, because

$$\frac{(1 - \bar{\alpha}_t)(2 - 2\bar{\alpha}_{t-1} - \sigma_t^2)\sigma_t^2}{(1 - \bar{\alpha}_{t-1})^2} \leq \frac{2\sigma_t^2(1 - \bar{\alpha}_t)}{1 - \bar{\alpha}_{t-1}}.$$

**Remark 2**   If we set $\sigma_t = \sqrt{1 - \bar{\alpha}_{t-1}}$, then equation 11 becomes

$$\tilde{\boldsymbol{x}}_t \leftarrow \tilde{\boldsymbol{x}}_t - \sqrt{1 - \bar{\alpha}_t}\tilde{\epsilon}(\tilde{\boldsymbol{x}}_t, \gamma S^*, C) + \sqrt{1 - \bar{\alpha}_t}\boldsymbol{z}, \quad \boldsymbol{z} \sim \mathcal{N}(\boldsymbol{0}, \mathbf{I})$$

which is equivalent to directly sampling $\tilde{\boldsymbol{x}}_t$ using equation 7 without sampling intermediate $\tilde{\boldsymbol{x}}_{t-1}$ in our Algorithm 1. Thus, directly sampling $\tilde{\boldsymbol{x}}_t$ using equation 7 is a special case of Fusion Sampling.

## 3   EXPERIMENTS

We conduct extensive experiments to compare the proposed framework with several baseline methods including Stable Diffusion[2] (Rombach et al., 2021), Textual Inversion (Gal et al., 2022), Dream-Booth (Ruiz et al., 2022) and E4T (Gal et al., 2023).

### 3.1   IMPLEMENTATION DETAILS

**Model Architecture**   Our PromptNet is initialized from the U-Net encoder of pre-trained Stable Diffusion 2. We add an extra convolutional layer so that the model can take both $\boldsymbol{x}_t$ and $\boldsymbol{x}_0$ as inputs. An extra convolutional layer is introduced for each intermediate layer in the backbone, which will map all intermediate features into the same dimensionality. Resulting features will be concatenated and projected into the text embedding space of Stable Diffusion 2 by a fully-connected layer.

**Pre-training Details**   We implement our framework on Nvidia A100 GPUs, based on Hugging Face Diffusers (von Platen et al., 2022). Our PromptNet is pre-trained for 300,000 steps on Conceptual Captions (CC3M) dataset (Sharma et al., 2018). Although CC3M contains 3 millions of image-text pairs, only image samples are used in training our PromptNet. We use AdamW (Loshchilov & Hutter, 2018) optimizer with learning rate of 1e-5 and batch size of 128. Data augmentation is not used during pre-training stage. Our pre-trained model will be publicly available upon acceptance.

**Fine-tuning Details**   Given a test image, PromptNet and all cross-attention layers of Stable Diffusion 2 are fine-tuned with a batch size of 16 and learning rate of 5e-5. Random affine transformation is applied as data augmentation, details are provided in the Appendix. 31 augmented data samples are generated for a given image, which only requires 0.7 second thus the computation cost can be ignored. Fine-tuning a model with ProFusion for 10 steps only needs 5 seconds on a single A100 GPU, indicating the efficiency of the proposed method, especially considering the impressive results we could obtain. In most experiments, we fine-tune our model for 50 steps for a given image unless specified. We also conduct ablation study with different fine-tuning steps, showing that slightly fine-tuning the model for a few seconds can already lead to good results.

### 3.2   MAIN RESULTS

Some generated results are shown in Figure 1, where only single image is provided to our model for customization. From the results, we can see that the proposed framework effectively achieves customized generation in arbitrary downstream domain. Our customized generation meets the specified text requirements while maintaining fine-grained details of the input image. More results are provided in the Appendix.

---

[2]Results of Stable Diffusion is obtained by directly feeding corresponding prompts into the model.

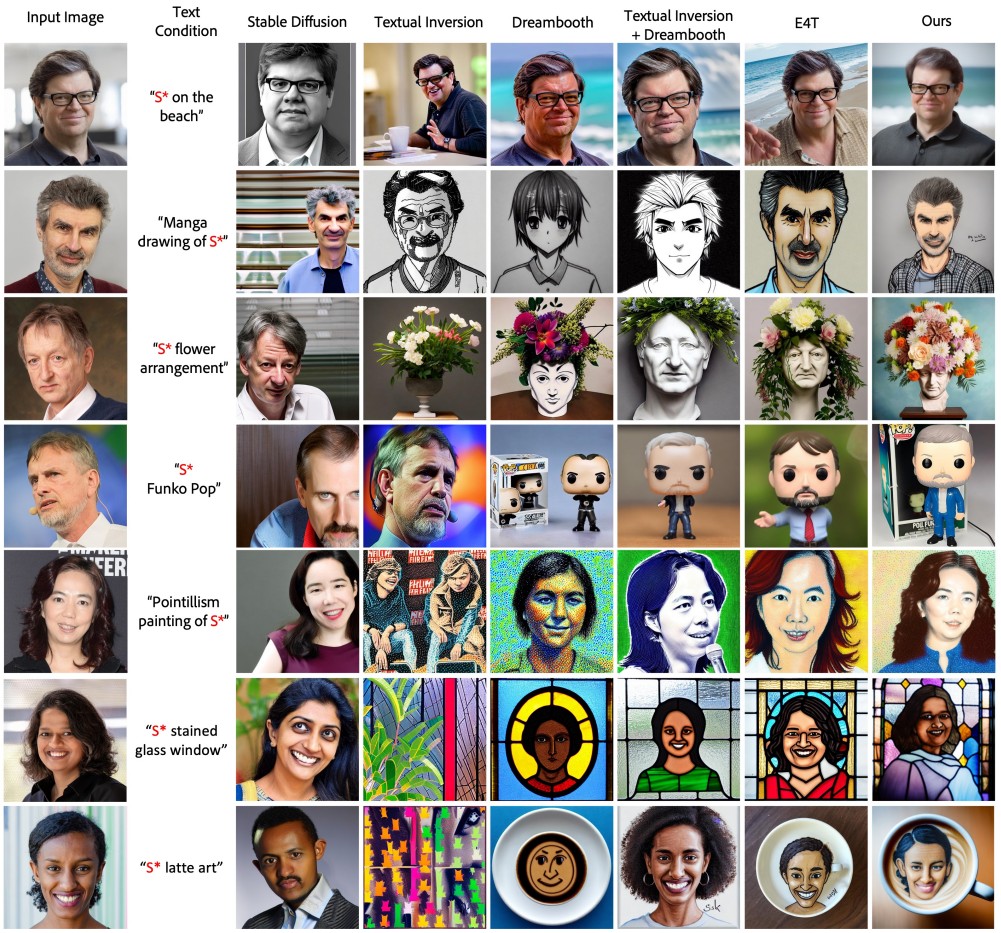

Figure 4: Comparison with baseline methods. Our proposed approach exhibits superior capability for preserving fine-grained details.

**Human Face Domain** Following Gal et al. (2023), we first evaluate our method on human face images provided in Gal et al. (2023). For simplicity, we fix some hyper-parameters including $\omega_1 = \omega_2 = 7.5$, $\sigma_t = \sqrt{(1 - \bar{\alpha}_{t-1})/(1 - \bar{\alpha}_t)}\sqrt{1 - \bar{\alpha}_t/\bar{\alpha}_{t-1}}$, and only tune $\gamma \in \{0.4, 0.5, 0.6, 0.7, 0.8\}$ during sampling. In general, larger $\gamma$ leads to better identity similarity at the expense of creativity. The qualitative comparison are presented in Figure 4, from which we can see that our framework results in better preservation of fine-grained details. Results of the related methods are directly taken from Gal et al. (2023).

Figure 5: Results on human face domain.

We also conduct quantitative evaluation. We use the prompts and images provided in Gal et al. (2023) and generate 10 images for each prompt-researcher combination. The generated images are fed into pre-trained CLIP models (Radford et al., 2021) to calculate the average image-prompt similarity[3]. We then calculate the identity similarity between generation and reference, which is the cosine similarity between features extracted by pre-trained face recognition models. The identity similarity is also averaged across different pre-trained models (Taigman et al., 2014; Schroff et al., 2015; Parkhi et al., 2015; Deng et al., 2019; Kim et al., 2022; Serengil & Ozpinar, 2020; 2021). The results are shown in Figure 5. Overall, our ProFusion performs better both in image-prompt and identity similarity, showing the effectiveness of the proposed method.

---

[3]The results are averaged over CLIP ViT-B/32, ViT-B/16, ViT-L/14, ViT-L/14@336px, RN101, RN50, RN50x4, RN50x16 and RN50x64.

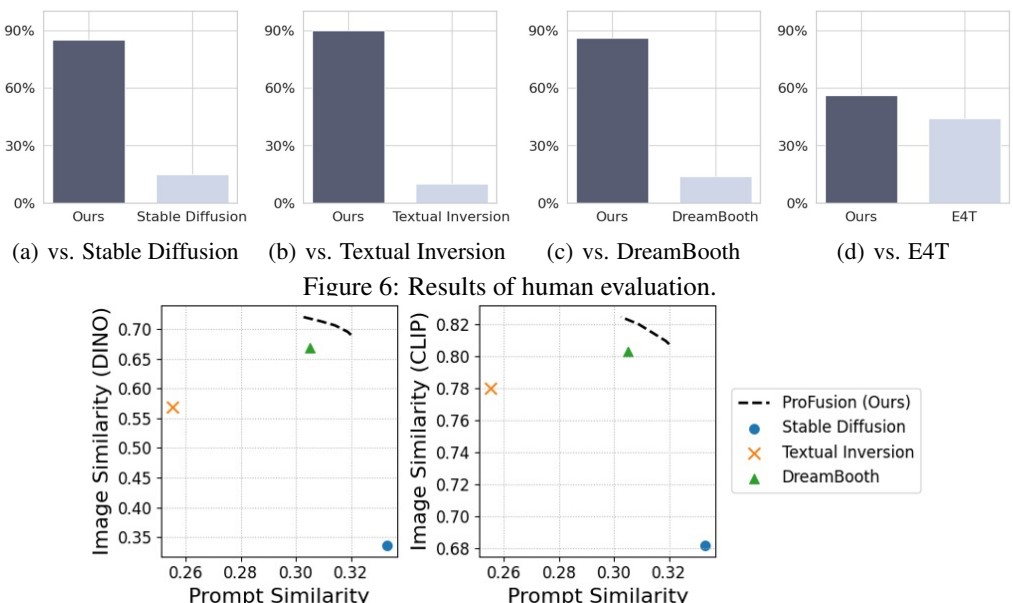

(a) vs. Stable Diffusion    (b) vs. Textual Inversion    (c) vs. DreamBooth    (d) vs. E4T

Figure 6: Results of human evaluation.

Figure 7: Results on DreamBooth benchmark.

**DreamBooth Benchmark** We conduct evaluation on the DreamBooth benchmark, which contains 30 subjects from different domains. 25 prompts are also provided for each subject. Following Ruiz et al. (2022), we generate 4 images for each prompt-subject combination. The generated images are used to compute the image-prompt similarity with CLIP, and the identity similarity with CLIP and DINO (Caron et al., 2021). The results are presented in Figure 7, along with some generated examples in Figure 8. As expected, our ProFusion obtains higher image-prompt similarity and image similarity than baselines, indicating better detail preservation and creativity.

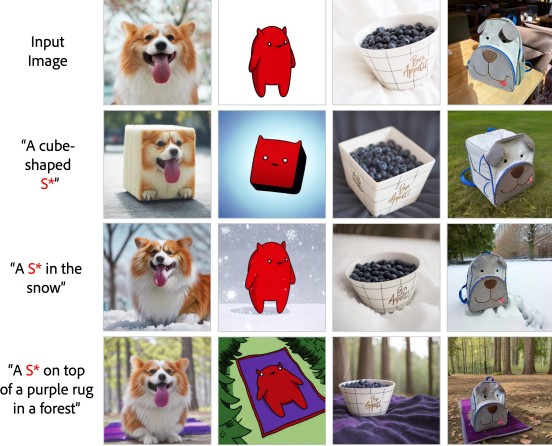

Figure 8: Generation on DreamBooth benchmark.

**Human Evaluation** We also conduct human evaluation with Amazon Mechanical Turk (MTurk). All workers are presented with two generated images from different methods along with original image and text requirements. Then they are asked to indicate their preferred choice. More details are provided in the Appendix. The results are shown in Figure 6, where we can find that our method obtains a higher preference rate compared to all other methods, indicating the effectiveness of our proposed framework.

### 3.3 ABLATION STUDY

We conduct several ablation studies to further investigate the proposed ProFusion.

**Fusion Sampling vs Baseline** First of all, we apply the proposed Fusion Sampling with both pre-trained and fine-tuned PromptNet. As shown in Figure 9, Fusion Sampling always leads to successful generation with respect to text condition, obtains better results than baseline classifier-free sampling. We then investigate the effects of removing fusion stage or refinement stage in the proposed Fusion Sampling. As we can see from Figure 12, removing refinement stage leads to the loss in detailed information, while removing fusion stage leads to a generated image with disorganized

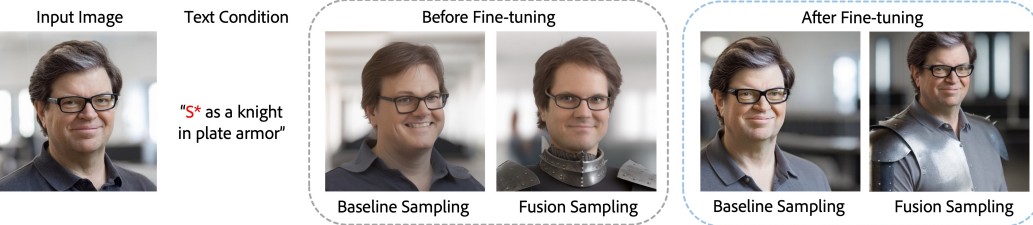

Figure 9: Fusion sampling always outperforms baseline classifier-free sampling because baseline can not solve overfitting problem thus fails to generate images aligned with text condition.

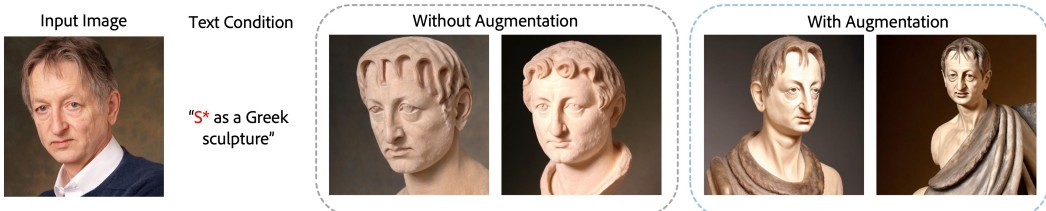

Figure 10: Data augmentation leads to better performance in terms of image fidelity and diversity.

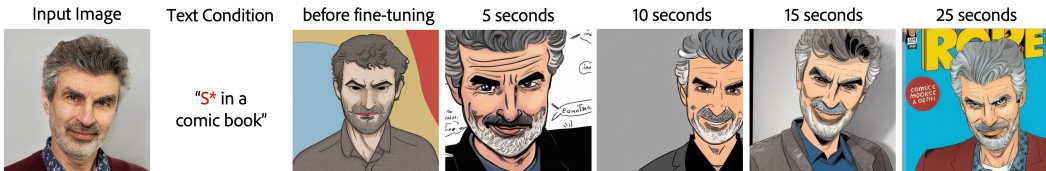

Figure 11: Results with different fine-tuning time.

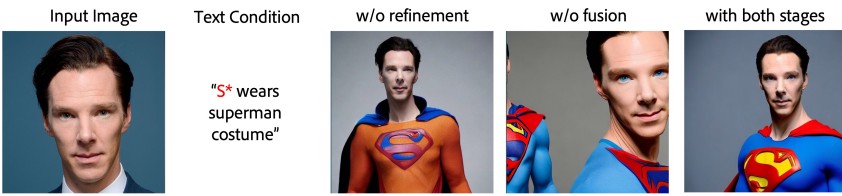

Figure 12: The impact of different stages in Fusion Sampling.

structure. Intuitively, $S^*$, which is the output of PromptNet, tries to generate a human face image following the structural information from the original image, while the text "is wearing superman costume" aims to generate a half-length photo. The conflicting nature of these two conditions results in an undesirable generation with a disorganized structure after we remove the fusion stage.

**Data Augmentation**   We then analyze the effects of data augmentation. In particular, we conduct separate fine-tuning experiments: one with data augmentation and one without, both models are tested with Fusion Sampling after fine-tuning. The results are shown in Figure 10, data augmentation leads to an performance improvement in terms of both generation fidelity and diversity.

**Fine-tuning Time**   We also conduct an ablation study to investigate the influence of fine-tuning time. Several models are fine-tuned then tested with Fusion Sampling, as shown in Figure 11. As expected, longer fine-tuning time leads to better performance. We also notice that reasonable result can actually be obtained with only 5 seconds of fine-tuning, which indicates the potential of ProFusion in real-world applications.

## 4   CONCLUSION

In this paper, we present ProFusion, a novel framework for customized text-to-image generation. Distinct from all previous methods that employ regularization, ProFusion successfully performs customized generation without any regularization, thus exhibits superior capability to preserve fine-grained detail with less training time. Extensive experiments have demonstrated the effectiveness of the proposed ProFusion.

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
