## A    MORE RESULTS

Some more generated examples are provided in Figure 13, Figure 14 and Figure 15.

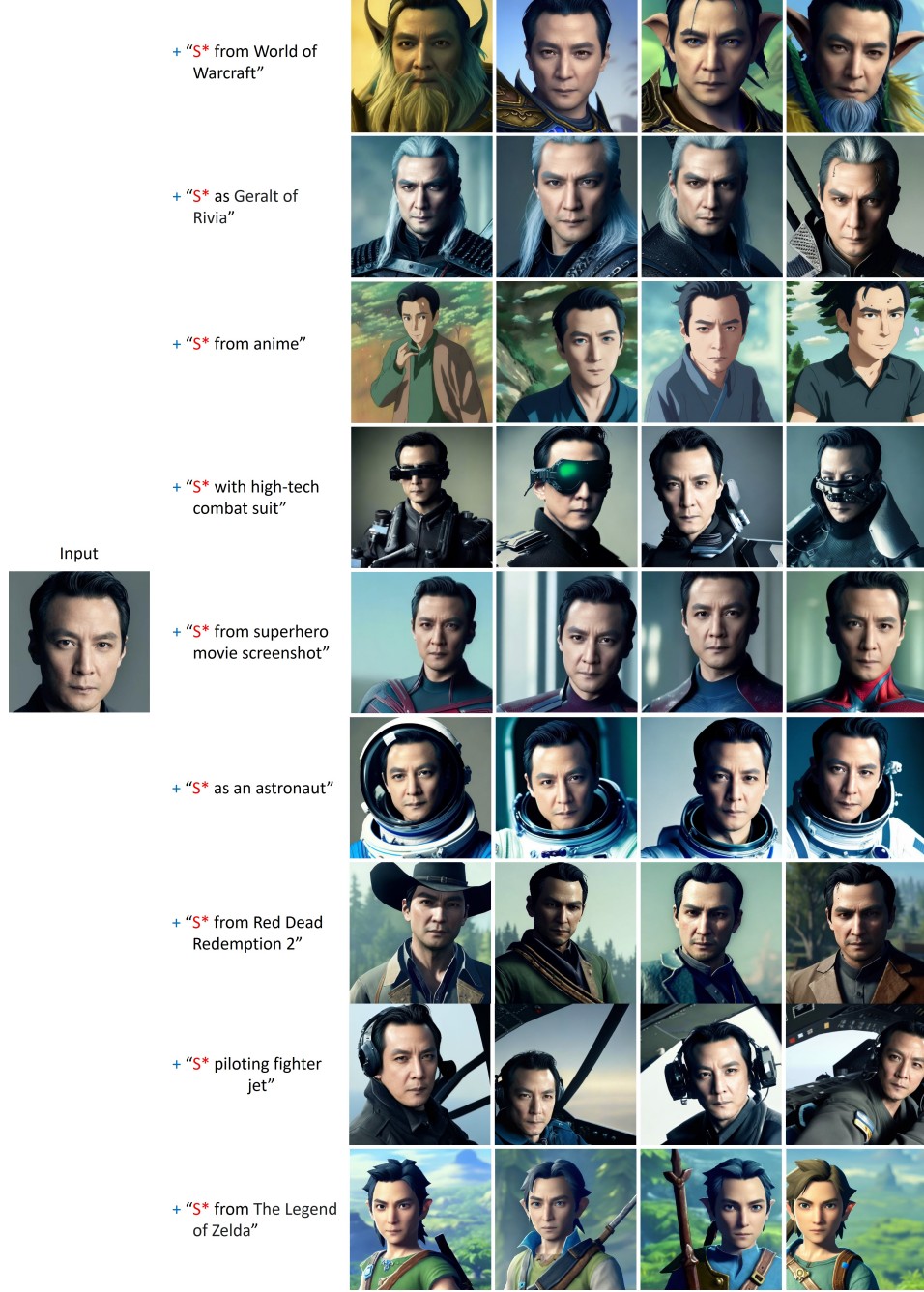

Figure 13: More results of customized generation with the proposed framework.

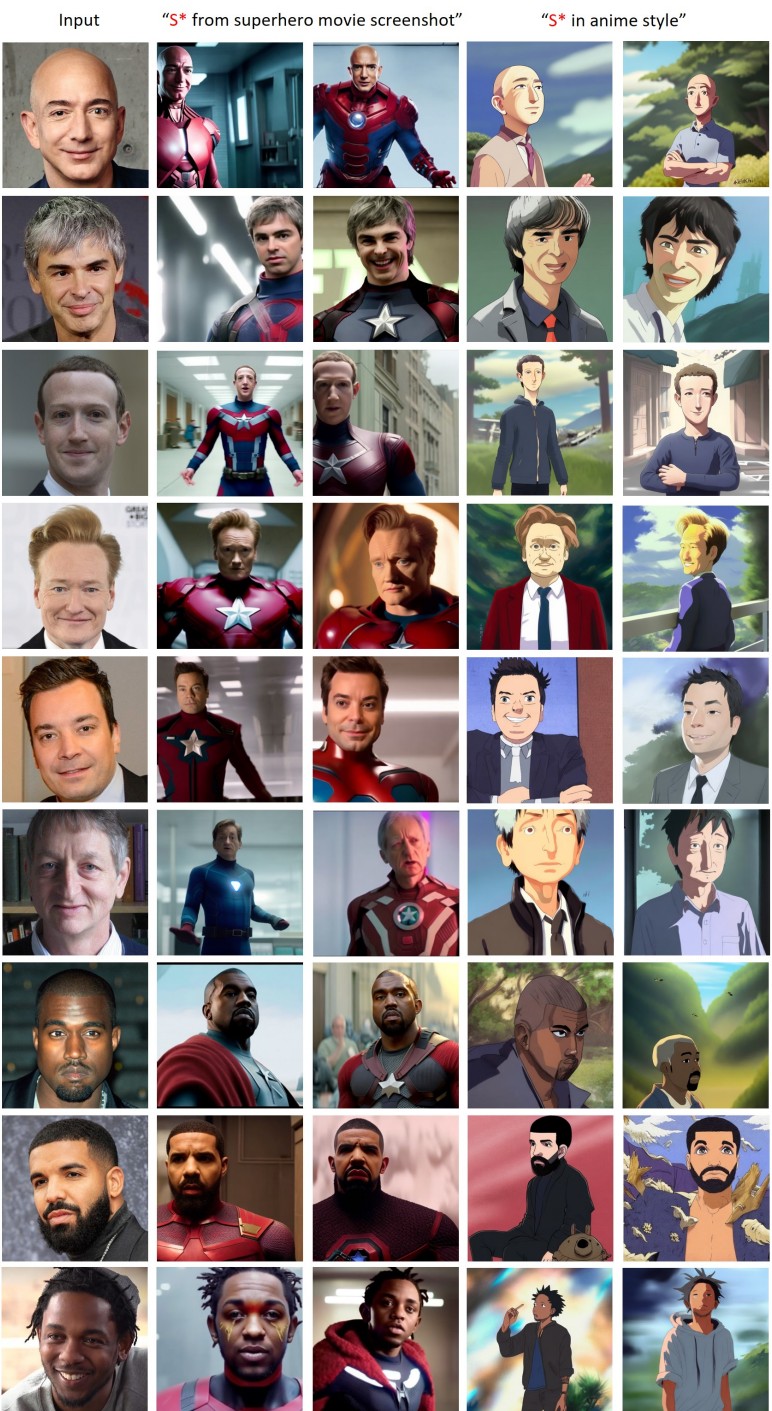

Figure 14: More results of customized generation with the proposed framework.

Input    "S* from superhero movie screenshot"    "S* in anime style"

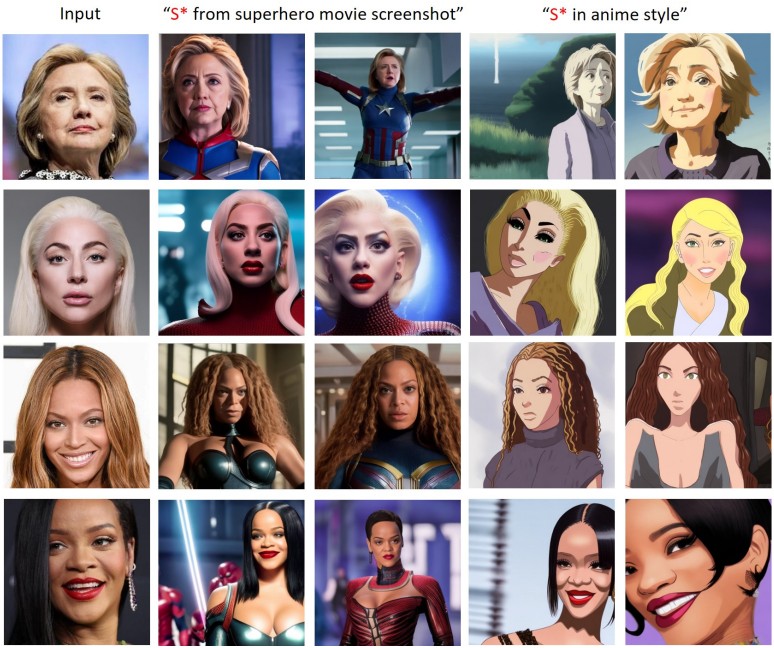

Figure 15: More results of customized generation with the proposed framework.

Input Image 1    "S* from superhero movie screenshot"    Input Image 2

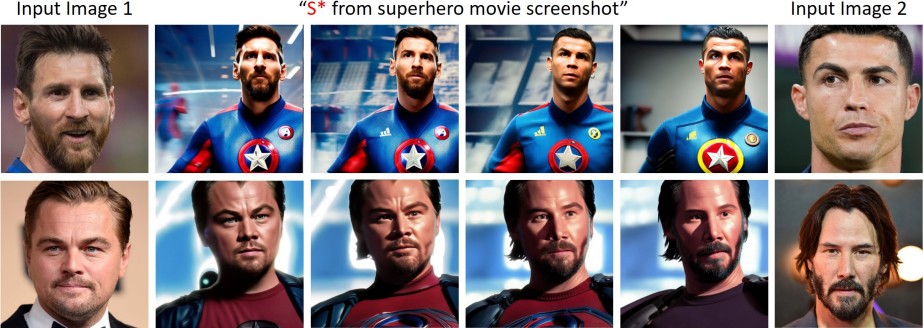

Figure 16: The proposed framework enables generation conditioned on multiple input images and text. Creative interpolation can be performed.

As mentioned previously, our proposed framework is also able to perform generation conditioned on multiple images. We also provide some examples in Figure 16.

We also conduct some ablation studies, investigating the proposed Fusion Sampling. Specifically, we test the influence of different $m$ in the fusion stage, which is shown in Figure 17. From the results we can see that $m = 1$ can already lead to promising results, increasing $m$ does not lead to obvious improvement. Considering the fact that increasing $m$ leads to longer inference time, we choose $m = 1$ in practice because of its promising performance and good efficiency.

We then conduct ablation study where different hyper-parameter in the Fusion Sampling is used. Specifically, we use different $\gamma$ in fusion stage, which is the simplest way to adjust the generation between focusing on identity preservation and focusing on creativity. The results are shown in Figure 18, from which we can conclude that larger $\gamma$ leads to better identity preservation with the loss of creativity; meanwhile, smaller $\gamma$ enables more creative generation, but it may also hurt the detail preservation.

We conduct experiment to see that whether the proposed framework can be used to capture style of the input image instead of object. As shown in Figure 19, the proposed method can also encode style into the embeddings, resulting in creative generation with the target style.

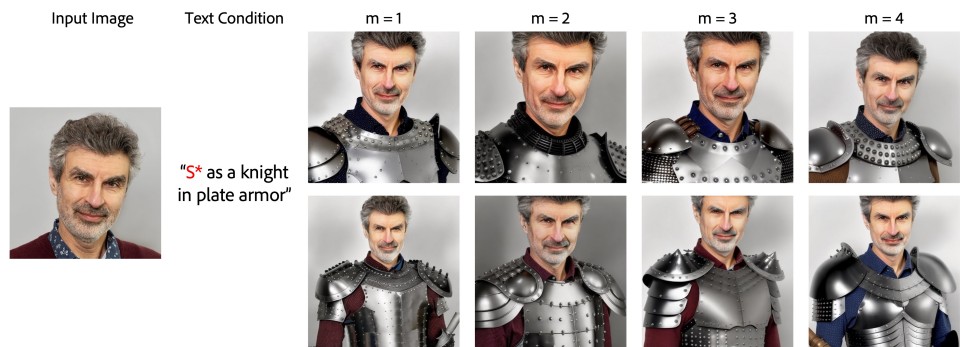

Figure 17: Ablation study of using different $m$ in the proposed Fusion Sampling.

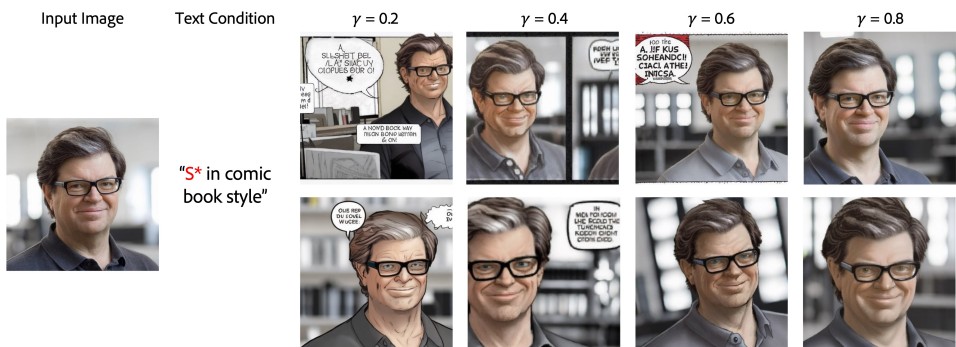

Figure 18: Ablation study of using different $\gamma$ in the proposed Fusion Sampling.

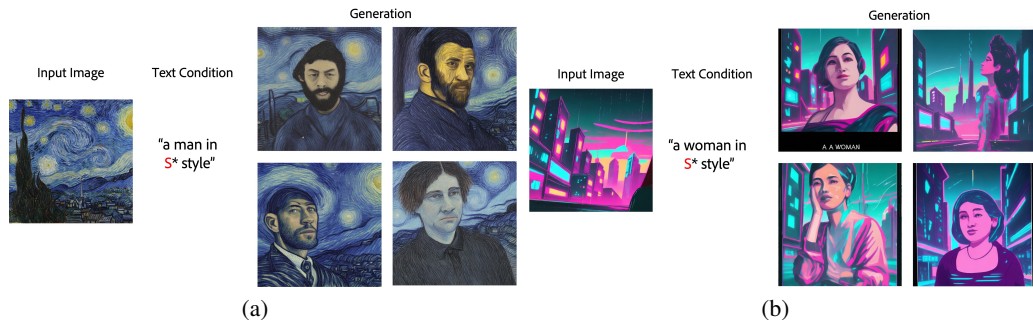

Figure 19: Style of the input image can also be captured by the proposed method.

To show the effectiveness of the proposed Fusion Sampling, we conduct an ablation study to compare Fusion Sampling with baseline sampling quantitatively. Specifically, we fine-tune the pre-trained model with different level of regularization where $\lambda \in [0, 0.001, 0.002, 0.005, 0.01, 0.02, 0.05, 0.1]$ and performed baseline sampling on the fine-tuned models. Data augmentation is also used for fair comparison. As shown in Figure 20, our Fusion Sampling always outperforms baseline sampling, under different level of regularization. We can conclude that tuning the hyper-parameter of regularization can not lead to better results than simply using the proposed regularization-free method.

Furthermore, from Figure 20 we can also see that changing the level of regularization will directly influence the identity similarity and prompt similarity as expected, which verifies our motivation of the proposed method.

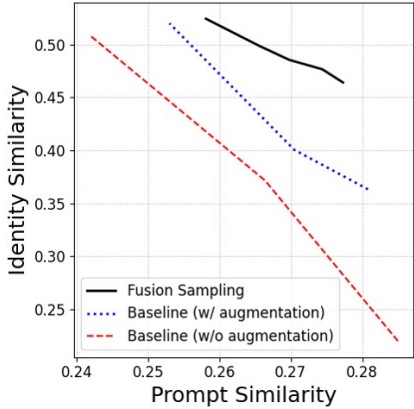

Figure 20: Comparison of proposed Fusion Sampling and baseline sampling.

## B  MORE IMPLEMENTATION DETAILS

Random affine transformation is used during fine-tuning to perform data augmentation. Specifically, the scaling factor is uniformly sampled from $[0.5, 1.0]$, rotation degree is uniformly sampled from $[-10°, 10°]$, random horizontal and vertical shift is applied at pixel-level with a value sampled from $[-100, 100]$.

**Human Evaluation**  Due to the fact that we do not have official implementation and pre-trained models of E4T Gal et al. (2023), we directly take some generated examples from their paper for fair comparison. Then we use corresponding prompts in our framework to generate images to be compared. Specifically, there are 39 source image and prompt pairs for five different methods and each generated image is evaluated by five different workers with expertise. These workers are all from the US and required to have performed at least 10,000 approved assignments with an approval rate $\geq 98\%$. The human evaluation user interface is shown in Figure 6.

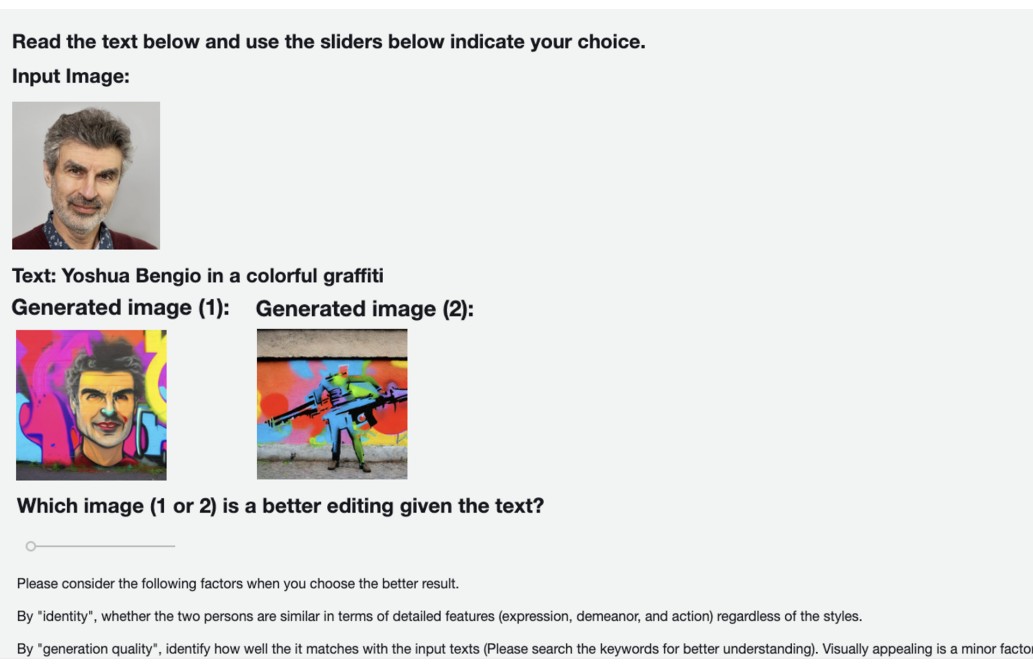

Figure 21: Human Evaluation User Interface.