# OpenReview forum: "Enhancing Detail Preservation for Customized Text-to-Image Generation: A Regularization-Free Approach"
_ICLR.cc/2024/Conference — Submitted to ICLR 2024_

### Official Review · Reviewer_qTeR · 2023-10-29

**Soundness:** 2 fair
**Presentation:** 2 fair
**Contribution:** 2 fair
**Rating:** 3
**Confidence:** 3

**Summary:**

The authors observe that the commonly used regularization to avoid overfitting in customized text-to-image generative models may lead to the loss of detailed information on the customized objects. The authors propose to balance the influences of the prompt condition and the customization condition S* instead of applying regularization to avoid overfitting, ensuring the preservation of customized details and flexibility to work with diverse prompts.

**Strengths:**

- Motivated by interesting observations
- Novel idea of removing regularization to preserve detailed information
- Computational resource-efficient approach

**Weaknesses:**

- Encoder-based customized Text-to-Image generation is not new and missing comparisons with important previous works: [1][2][3][4]
- The important term "independent conditions" is not clearly defined.
- Figure 9 shows that before fine-tuning, fusion sampling even leads to worse identity-preserving performance than baseline sampling, implying that performance gain in identity similarity mainly comes from fine-tuning.
- In Figure 4, the proposed method does not present visually better results compared to E4T.


[1] Yuxiang Wei, Yabo Zhang, Zhilong Ji, Jinfeng Bai, Lei Zhang, and Wangmeng Zuo. ELITE: Encoding visual concepts into textual embeddings for customized text-to-image generation. arXiv preprint arXiv:2302.13848, 2023.
[2] Chen, Wenhu and Hu, Hexiang and Li, Yandong and Ruiz, Nataniel and Jia, Xuhui and Chang, Ming-Wei and Cohen, William W. Subject-driven Text-to-Image Generation via Apprenticeship Learning. NeurIPS 2023.
[3] Rinon Gal, Moab Arar, Yuval Atzmon, Amit H. Bermano, Gal Chechik, Daniel Cohen-Or. Encoder-based Domain Tuning for Fast Personalization of Text-to-Image Models. arXiv preprint arXiv:2302.12228 (2023).
[4] Xuhui Jia, Yang Zhao, Kelvin C.K. Chan, Yandong Li, Han Zhang, Boqing Gong, Tingbo Hou, Huisheng Wang, Yu-Chuan Su. Taming Encoder for Zero Fine-tuning Image Customization with Text-to-Image Diffusion Models. arXiv preprint arXiv:2304.02642, 2023.

**Questions:**

- The authors aim at "detail preservation" but do not clearly explain what information these details include. What is the difference between detail preservation and identity preservation?
- Minor issue of presentation. Figures 6 and 7 on page 8 are squeezed and Figure 7 has slightly occluded the caption of Figure 6. May rearrange for better visualization.

---

> ### Author Response · Authors · 2023-11-10
> **Response to Reviewer qTeR**
>
> We thank the reviewer for the helpful reviews, below we address some questions and concerns.
>
> Q: Some related works are missing:
>
> A: Actually, [3]  (E4T) is the most important baseline we have already discussed in our paper, our method obtained better results as presented in the paper. We will include [1] and [4], and add comparison. Although [2] is not an encoder-based method, we will add a discussion.
>
> Q: The important term "independent conditions" is not clearly defined.
>
> A: By "independent conditions", we simply mean the two conditions $S^*$ and $C$ are independent given the generation, i.e, $p(S^*,C|x) = p(S^*| x)p(C|x)$.
>
> Q: Question about identity preservation.
>
> A: We want to emphasize that we never claimed the better fine-grained details can be obtained by a sampling method. What we claimed is that **details can be better preserved if the model is trained/fine-tuned without regularization**. However, **baseline sampling fails to generate desired images with respect to text** in this case. Figure 9 shows that Fusion Sampling can generate images aligned with text, under the setting without regularization, no matter before or after fine-tuning. Better details are certainly obtained by fine-tuning. However, without Fusion Sampling, we can not enjoy the setting without regularization which has the best detail preservation.
>
> That's why we propose to **fine-tune without regularization and perform Fusion Sampling after fine-tuning**. As we show in the paper, compared to related encoder-based method [3] which **fine-tunes the model with regularization and performs baseline sampling**, better results are obtained by our method.
>
> Furthermore, we are able to perform more flexible generation with Fusion Sampling: we can choose to emphasize either better creativity or better details, which is why the results of our method in Figure 5 and Figure 7 are represented by line instead of point like other methods. And our line is above all the points, indicating better results are obtained by our method.
>
> Q: About experiment results.
>
> A: Customized generation is subjective and hard to evaluate. That's why we performed both quantitative evaluation and human evaluation. In quantitative evaluation, pre-trained models are used to compare extracted features from original and generated images. Better quantitative results are obtained by our method. In human evaluation, workers from Amazon Mechanical Turk are asked to compare results from different methods. All the workers have performed at least 10,000 approved assignments with an approval rate ≥ 98%, thus we believe the human evaluation results from the workers can objectively illustrate the effectiveness of our proposed method.
>
>
>
> [1] Yuxiang Wei, Yabo Zhang, Zhilong Ji, Jinfeng Bai, Lei Zhang, and Wangmeng Zuo. ELITE: Encoding visual concepts into textual embeddings for customized text-to-image generation. arXiv preprint arXiv:2302.13848, 2023.
>
> [2] Chen, Wenhu and Hu, Hexiang and Li, Yandong and Ruiz, Nataniel and Jia, Xuhui and Chang, Ming-Wei and Cohen, William W. Subject-driven Text-to-Image Generation via Apprenticeship Learning. NeurIPS 2023.
>
> [3] Rinon Gal, Moab Arar, Yuval Atzmon, Amit H. Bermano, Gal Chechik, Daniel Cohen-Or. Encoder-based Domain Tuning for Fast Personalization of Text-to-Image Models. arXiv preprint arXiv:2302.12228 (2023).
>
> [4] Xuhui Jia, Yang Zhao, Kelvin C.K. Chan, Yandong Li, Han Zhang, Boqing Gong, Tingbo Hou, Huisheng Wang, Yu-Chuan Su. Taming Encoder for Zero Fine-tuning Image Customization with Text-to-Image Diffusion Models. arXiv preprint arXiv:2304.02642, 2023.

---

> > ### Comment · Reviewer_qTeR · 2023-11-21
> >
> > Thank the authors for the response. However, the response does not fully address my concerns. For example:
> >
> > - Fusion sampling is an important part of the proposed method, but its effectiveness is only demonstrated through one qualitative example in Fig.9.
> > - I agree with other reviewers that the tradeoff between detail preservation and text alignment and its relation with regularization needs to be more carefully analyzed to prove the motivation of the regularization-free approach.
> > - Why do the authors think suti [1] is not an encoder-based method?
> >
> > [1] Chen, Wenhu and Hu, Hexiang and Li, Yandong and Ruiz, Nataniel and Jia, Xuhui and Chang, Ming-Wei and Cohen, William W. Subject-driven Text-to-Image Generation via Apprenticeship Learning. NeurIPS 2023.

---

> ### Author Response · Authors · 2023-11-21
> **Response to Reviewer qTeR**
>
> We will add more experiment results to illustrate the effectiveness of the Fusion Sampling in the Appendix soon.
>
> There might be some misunderstanding, we would like to clarify more on the "encoder-based" method.
> SuTI directly adopts the architecture from Re-Imagen [1], which is a retrieval augmented diffusion model. Different from E4T, ELITE or our method, it does not introduce an extra encoder network outside the diffusion model, but directly use the "encoder part" of the UNet. We did not call it encoder-based method, just like we would not call diffusion model itself as an encoder-based method.
> Furthermore, SuTI focuses on apprenticeship learning so that the final model can imitates the behaviors of different experts, rather than focusing on design encoder mapping to map input image into embeddings like E4T do. We are happy to add SuTI as related work for discussion, but want to emphasize the difference between it and other works like E4T and ELITE.
>
> [1]. Re-Imagen: Retrieval-Augmented Text-to-Image Generator. Wenhu Chen, Hexiang Hu, Chitwan Saharia, William W. Cohen.

---

### Official Review · Reviewer_GgLb · 2023-10-29

**Soundness:** 3 good
**Presentation:** 3 good
**Contribution:** 3 good
**Rating:** 6
**Confidence:** 3

**Summary:**

This paper proposes a regularization-free and detail-preservation approach for customized text-to-image generation.
To this end, this paper introduces PropFusion to tackle over-fitting problem without the widely used regularization.
Therefore, PropFusion significantly reduces training time while achieve enhanced preservation of fine-grained details.
Moreover, it also introduces a novel sampling method namely Fusion Sampling to meet the requirements of text prompts.
Extensive experiments demonstrate the superiority of the proposed method.

**Strengths:**

See summary.

**Weaknesses:**

1. Can the authors elaborate more derivation details from Eq.10 to Eq.11?
2. Comparisons with IP-Adapter[1]. IP-Adapter also projects the input image into the text embedding space, but requires no additional finetuning or well-designed sampling. The authors are encouraged to compare to this simple baseline.

[1] Hu Ye, Jun Zhang, Sibo Liu, Xiao Han, Wei Yang. IP-Adapter: Text Compatible Image Prompt Adapter for Text-to-Image Diffusion Models.

**Questions:**

See weaknesses.

---

> ### Author Response · Authors · 2023-11-21
> **Response to Reviewer GgLb**
>
> We thank the reviewer for the insightful comments and suggestions, below we address some concerns and answer some questions.
>
> Q: From Equation 10 to Equation 11.
>
> A: Equation 11 is derived from Equation 8 and Equation 10. In Equation 8, we obtain an inaccurate prediction for timestep $t-1$, which is denoted as $\widetilde{x_{t-1}}$. In equation 10, we inject the structure information in $\widetilde{x_{t-1}}$ into the current noised sample $x_t$ to obtain an updated sample $x_t$.
> As we discussed in the paper. Simply feeding the conditions independently into the model will lead to structure conflict. Thus we need to first obtain an “inaccurate” next-step sample, which is harmonized in terms of structure information . It is called “inaccurate” because which it is conditioned on $\gamma S^*$. Using this $\widetilde{x_{t-1}}$ as a prediction for $x_{t-1}$ leads to generation with some details lost, as shown in Figure 12. But this “inaccurate” sample  $\widetilde{x_{t-1}}$ does not have structure conflict anymore. Thus we propose to use its structure to infer a corresponding updated sample  $\widetilde{x_{t}}$ for current time step $t$, then perform sampling based on this  $\widetilde{x_{t}}$ instead of  $x_t$.
>
> Q: Related works.
>
> A: Thanks for the suggestion, we will add the discussion.

---

> > ### Comment · Reviewer_GgLb · 2023-11-23
> >
> > Thanks for your elaborate response and I will keep my score 6 to accept this paper.

---

### Official Review · Reviewer_KZRC · 2023-10-31

**Soundness:** 2 fair
**Presentation:** 3 good
**Contribution:** 2 fair
**Rating:** 5
**Confidence:** 3

**Summary:**

This paper proposes a novel framework called ProFusion to tackle the over-fitting problem in customized text-to-image with an encoder network and a novel sampling method. The encoder network receives an image depicting a customized object, which makes the generation condition on that object. Then, a novel sampling method called Fusion sampling is proposed to enhance the generation to be conditioned on both the input image and the arbitrary user input text. The experiments conducted demonstrate the effectiveness and superiority of the proposed framework.

**Strengths:**

1. The paper is well-organized, and the method is easy to follow. The core contribution Fusion sampling is novel and may have a broad impact on the conditional image generation community.
2. The most important contribution of the paper is that it proposes a framework without regularization techniques to prevent over-fitting in customized text-to-image generation. The encoder network, which converts the image containing the customized object into an embedding $S^*$, makes the diffusion model condition on that object. Unlike previous methods adopting regularization techniques to prevent over-fitting, this paper proposes a novel Profusion sampling method. By assuming $S^*$ and the arbitrary user input text $C$ are independent, the sampling method decomposes the noise prediction into two terms that take both $S^*$ and $C$ into consideration.
3. The experiments show that the proposed method achieves superior capability for preserving fine-grained details.

**Weaknesses:**

1. Line#5 to Line#7 in Algorithm 1 is the standard diffusion denoising sampling step. However, the intuition behind #Line9 in the algorithm is unclear.
2. The corresponding ablation study showing the difference between using or not using the two stages in the algorithm is not convincing enough.

**Questions:**

1. Please clarify the intuition or motivation behind Line#9 in the fusion stage in the fusion sampling method.
2. Please provide the corresponding ablation studies to support that using m=1 in practice works well for the fusion sampling method.
3. Do the baseline methods all use the data augmentation method in the paper? If not, the comparison may be unfair.
4. It is better to report the exact training time, fine-tuning time, and the GPU devices used for reproduction consideration.
5. Is it possible to combine the proposed sampling method with other regularization techniques to further improve the performance? Or is it possible to adopt the sampling method to other methods to verify its effectiveness in improving detail preservation?
6. One may trade off the detail preservation for more creative generations or the other way around. Can the authors provide such trade-offs in the proposed framework?
6. Is the method capable of capturing the style of the input customized image? For instance, can the method perform style transfer like generating an image by the prompt "a car in the style of $S^*$".

---

> ### Author Response · Authors · 2023-11-21
> **Response to Reviewer KZRC**
>
> We thank the reviewer for the insightful comments and suggestions, below we address some concerns and answer some questions.
>
> Q: About line#9 in the algorithm.
>
> A: As we discussed in the paper. Simply feeding the conditions independently into the model will lead to structure conflict. Thus we need to first obtain an “inaccurate” next-step sample, which is harmonized in terms of structure information . It is called “inaccurate” because which it is conditioned on $\gamma S^*$. Using this $\widetilde{x_{t-1}}$  as a prediction for $x_{t-1}$ leads to generation with some details lost, as shown in Figure 12. But this “inaccurate” sample  $\widetilde{x_{t-1}}$  does not have structure conflict anymore. Thus in line #9 of the algorithm we propose to use its structure to infer a corresponding updated sample  $\widetilde{x_{t}}$  for current time step $t$, then perform sampling based on this  $\widetilde{x_t}$  instead of  $x_t$.
>
> Q: About m=1 in the algorithm.
>
> A: In all the experiments reported in the paper, we use m=1, which can already lead to better results than other methods. We believe that it is enough to show that m=1 works well in practice. We add ablation study on m>1, in the revised Supplementary Material. From which we can see that increasing m does not lead to noticeable improvement. Considering the fact that increasing m will lead to longer sampling time, thus we believe using m=1 in practice is a better option.
>
>
> Q: Do all the baseline methods use data augmentation?
>
> A: We didn’t use data augmentation in the pre-training stage, which is the same as other method. Data augmentation during fine-tuning is part of our proposed method. We will provide ablation study of comparing baseline sampling and Fusion Sampling, under different settings including with data augmentation, without data augmentation, and different level of regularization during fine-tuning.
>
> Q: Implementation details.
>
> A: We conduct all the experiments on Nvidia A100 GPUs. The pre-training time for encoder on CC3M dataset cost around 1 week on 8 A100 GPUs. The fine-tuning time on a single A100 GPU is around 25 seconds for 50 steps.
>
> Q: Is it possible to combine the proposed method with other regularization methods?
>
> A: Good question. Yes, the proposed method can be combined with any regularization methods as the Fusion Sampling will not influence the training, it only changes the inference. However, whether it can lead to improvement over a model without regularization depends on the regularization method itself. Currently, applying Fusion Sampling on a model with L2 regularization does not lead to observable improvement. Because some fine-grained details will be lost by applying regularization, and can not be recovered by simply using Fusion Sampling.
>
> Q: Trade-off between creativity and detail preservation.
>
> A: Good question, we provide some results in the revised Supplementary Material. We can easily control the generation, obtaining different level of creativity and detail preservation, by tuning a single hyper-parameter $\gamma$.
>
> Q: Can the method capture style?
>
> A: Good question, we provide some results in the updated Supplementary Material, from which we can see that the proposed method can successfully capture the style of the input image and combine it with user-input text to create new generations.

---

> ### Comment · Reviewer_KZRC · 2023-11-22
> **Response to the rebuttal by Reviewer KZRC**
>
> I thank the authors for the rebuttal. The response addresses most of my concerns. However, I agree with other reviewers that the motivation of the need for "regularization-free" is not comprehensively analyzed. Perhaps the authors should emphasize more on the problem of overfitting to $S^*$ instead of detail preservation. Also, the effectiveness of the Fusion sampling method has not been demonstrated convincingly in the current version. Therefore, I lowered my rating accordingly.

---

### Official Review · Reviewer_Y2oZ · 2023-11-04

**Soundness:** 2 fair
**Presentation:** 2 fair
**Contribution:** 2 fair
**Rating:** 5
**Confidence:** 4

**Summary:**

This paper tackles the task of customized text-to-image generation. It trains the PromptNet to map concepts in an input image into a text embedding for subsequent text-to-image generation. During training, the input image is augmented with affine transformation, and the UNet attention layers are fine-tuned. During sampling, the proposed Fusion Sampling method separates the obtained prompt S* and the arbitrary text prompt C, and feed them into UNet separately to avoid S* overriding C.

**Strengths:**

1. The paper presentation is clear, and easy to follow.
2. This paper presents some nice results of cutomized text-to-image generation.

**Weaknesses:**

1. The motivation of the need for "regularization-free" is not comprehensively demonstrated. See Question No.1 for more details.
2. The contribution of PromptNet is limited, where the main difference compared to other encoder-based approaches (e.g. ELITE) is that the affine transformation is introduced.
3. Similar ideas towards Fusion Sampling has been seen in several works [1][2], where the input condition is decomposed into multiple segments, and fed into the diffusion model separately.
[1] Compositional Visual Generation with Composable Diffusion Models (ECCV 2022)
[2] Collaborative Diffusion for Multi-Modal Face Generation and Editing (CVPR 2023)

**Questions:**

1. The motivation is that "regularization can result in detail lost". However, the validity of this claim depends on the degree of regularization applied, and the degree of detail loss suffered. In figure 3, what is the regularization weighting (i.e. lambda)? Have you tried an even smaller regularization weighting, and what is the degree of detail lost then?
2. In section 2.1, the observation is made that "without regularization, the UNet has a tendency to overly prioritize the S* concept, and overshadowing C". The question arises as to why feeding C independently into the UNet can address this issue, since the text-to-image mapping is not altered by the FusionSampling strategy.

---

> ### Author Response · Authors · 2023-11-21
> **Response to Reviewer Y2oZ**
>
> We thank the reviewer for the insightful comments and suggestions, below we address some concerns and answer some questions.
>
> Q: About regularization used in verifying the claim "regularization can result in detail lost".
>
> A: In experiments, we tried regularization $\lambda \Vert S \Vert^2$ with $\lambda$ selected from [1e-5, 1e-4, 1e-3, 1e-2, 1]. Figure 3 is presented to illustrate the influence of regularization on generation, small regularization like 1e-5 will result in failure in generating creative contents, while larger regularization will lead to identity loss. In customization, we are looking for a point where the model can preserve the details while having the capability to generate creative contents. However, simply tuning regularization hyper-parameter can not solve this problem.
>
> Q: Why feeding C independently into the UNet can address this issue.
>
> A: When we feed both $S^*$ and $C$ into the model, the model will ignore $C$ because $S^*$ is too strong. But if we feed $C$ along to the model, for sure the model will notice $C$, as there is no other condition here. What’s more, feeding conditions independently gives us the flexibility to control the influence of each condition, through a hyper-parameter $w_i$ in classifier-free guidance. We can control how much should the results be conditioned on $S^*$ and $C$.
> As we show in the paper. Feeding conditions independently can not fully solve the problem, that’s why we need another fusion stage, to avoid some structure conflict in the generated results.

---

> > ### Comment · Reviewer_Y2oZ · 2023-11-22
> >
> > I appreciate the authors for their response.
> >
> > I can understand your claim that "it's difficult to find a regularisation weighting to ensure both detail preservation and content diversity". I just don't find the experimental proof convincing.
> > Figure 3 only shows 3 different degrees of regularisation, and how do they match to the 4 weightings [1e-5, 1e-4, 1e-3, 1e-2, 1]? Furthermore, in the two rows of Figure 3, the smallest regularisation image (leftmost) in both rows look almost identical to me, but their text conditions are different. I am not sure whether this is the result caused by 1e-5 regularisation?
> >
> > It would be good if there are further explanations on why S* can be too strong. However, most my concerns in the second question have been addressed.

---

### Comment · Area_Chair_riBj · 2023-11-20

Dear reviewers,

As the Author-Reviewer discussion period is going to end soon, please take a moment to review the response from the authors and discuss any further questions or concerns you may have.

Even if you have no concerns, it would be helpful if you could acknowledge that you have read the response and provide feedback on it.

Thanks,
AC

---

### Author Response · Authors · 2023-11-23
**Updated Results**

We thank all the reviewers for their insightful comments and suggestions. We have updated some new results in the Appendix, including:

1. A quantitative comparison between baseline sampling and the proposed Fusion Sampling. Specifically, different models are obtained corresponding to level of regularization is applied with $\lambda \in \left[ 0, 0.001, 0.002, 0.005, 0.01, 0.02, 0.05, 0.1  \right]$, data augmentation is also applied for fair comparison. Fusion Sampling outperforms all the results of baseline sampling, showing the effectiveness of the proposed method, indicating that tuning the hyper-parameter of regularization can not lead to better results than the proposed regularization-free method;

2. We provide comparison of using different $m$ in the proposed algorithm, from which we can see that $m=1$ can already lead to good results, and there is no noticeable improvement when larger $m$ is used. Considering that increasing $m$ leads to longer inference time, we believe $m=1$ is a good choice in practice;

3. We provide qualitative results of using different $\gamma$ in proposed Fusion Sampling, showing that we have the flexibility to control the generation by simply tuning a single parameter: larger $\gamma$ leads to better identity similarity, smaller $\gamma$ leads to better creativity;

4. We also provide qualitative results, showing that the proposed method can not only encode object or person into the embedding $S^*$, but also encode the style of the original image, which can further lead to creative generation that aligns with the target style and input text;

---

### Meta-Review · Area_Chair_riBj · 2023-12-05

**Metareview:**

This paper proposes ProFusion for customized text-to-image generation. The key of the proposed approach is Fusion sampling, where guidance from different conditions (i.e., subject embedding and text embedding) are separated. The reviewers think that the technical novelty is limited, motivation is not clearly verified, and comparison to related works is lacking. After going through the paper, reviewers, and response, the AC agrees with the reviewers regarding the above concerns. Therefore, a rejection is recommended.

**Justification For Why Not Higher Score:**

The reviewers reached a consensus of rejecting this paper, and the AC agrees with the concerns raised by the reviewers. Therefore, the AC recommends a reject.

**Justification For Why Not Lower Score:**

N/A

---

### Decision · Program_Chairs · 2024-01-16

Reject